# Fatigue Crack Evaluation with the Guided Wave–Convolutional Neural Network Ensemble and Differential Wavelet Spectrogram

**DOI:** 10.3390/s22010307

**Published:** 2021-12-31

**Authors:** Jian Chen, Wenyang Wu, Yuanqiang Ren, Shenfang Yuan

**Affiliations:** Research Center of Structural Health Monitoring and Prognosis, State Key Laboratory of Mechanics and Control of Mechanical Structures, Nanjing University of Aeronautics and Astronautics, No. 29 Yudao Street, Nanjing 210016, China; cj1108@nuaa.edu.cn (J.C.); wwenyang@nuaa.edu.cn (W.W.); renyuanqiang@nuaa.edu.cn (Y.R.)

**Keywords:** fatigue crack evaluation, guided wave, convolutional neural network ensemble, time-frequency spectrogram

## Abstract

On-line fatigue crack evaluation is crucial for ensuring the structural safety and reducing the maintenance costs of safety-critical systems. Among structural health monitoring (SHM), guided wave (GW)-based SHM has been deemed as one of the most promising techniques. However, the traditional damage index-based method and machine learning methods require manual processing and selection of GW features, which depend highly on expert knowledge and are easily affected by complicated uncertainties. Therefore, this paper proposes a fatigue crack evaluation framework with the GW–convolutional neural network (CNN) ensemble and differential wavelet spectrogram. The differential time–frequency spectrogram between the baseline signal and the monitoring signal is processed as the CNN input with the complex Gaussian wavelet transform. Then, an ensemble of CNNs is trained to jointly determine the crack length. Real fatigue tests on complex lap joint structures were carried out to validate the proposed method, in which several structures were tested preliminarily for collecting the training dataset and a new structure was adopted for testing. The root mean square error of the training dataset is 1.4 mm. Besides, the root mean square error of the evaluated crack length in the testing lap joint structure was 1.7 mm, showing the effectiveness of the proposed method.

## 1. Introduction

Structural integrity is a key issue for safety-critical systems such as aircraft, infrastructures, and nuclear plants [1,2]. Commonly, fatigue cracks are one of the primary causes of structural failure [3,4]. It is crucial to examine the state of fatigue cracks in the structure to demine whether or when maintenance operations are needed. In recent decades, structural health monitoring (SHM) [5] has shown great potential in interrogating structural health state in real-time, whose results can be incorporated for the timely optimizing of the maintenance strategies, consequently ensuring structural safety and reducing the maintenance cost [6].

In the SHM field, different kinds of methods have been developed [7,8,9,10]. Among them, the guided wave (GW)-based SHM [11,12,13] has been deemed as one of the most promising techniques due to its sensitivity to small damages and capability of monitoring a region that is sometimes inaccessible [14]. In the GW-based method, the GW signal is excited to propagate in the structure. GW signals tuned by the crack are then collected and processed to evaluate the crack state. The basic method is to evaluate the scalar called the damage index [11] from GW signals in the time or frequency domain. This method needs a reasonable selection of the damage index and the wave packet for calculation. However, it would get into trouble when the wave packet is badly overlapped due to frequency dispersion and reflection from complex geometries in the structure [15]. Moreover, fatigue crack growth is a complicated process affected by uncertainties from sources like fatigue loading, boundaries, and environmental factors. These uncertainties would also cause changes to fatigue crack growth itself and GW signals, introducing difficulties for reliable crack evaluation. To deal with these problems, machine learning methods are adopted, such as the auto-regression model [16], the Gaussian mixture model [17], artificial neural networks [18], and hidden Markov models [19,20]. These methods are widely explored in the SHM field; however, most of them still require manual processing and the selection of GW damage index, which rely heavily on expert knowledge.

Recently, deep learning has already made a huge impact in areas such as cancer diagnosis, precision medicine, self-driving cars, predictive forecasting, and speech recognition [21]. More and more attention is paid to deep learning-based diagnosis due to its strong feature extraction and features fusion capability [22], especially for the fault diagnosis of rotary machines [23,24,25] and vision-based crack detection in infrastructures [26,27]. Moreover, attempts were made for applying deep learning in the GW-based SHM. For example, Xu et al. [28] organized GW damage indexes from different excitation–sensing paths as a one-dimensional vector, which was input into a convolutional neural network (CNN) for classifying fatigue crack levels in a lug structure. Rai et al. [29] and Mariani et al. [30] directly adopted the one-dimensional GW signal as the CNN input to localize and evaluate the notched damage in the plate structure. Lim et al. [31] repeated the same GW signal as a matrix, in which the CNN is used for monitoring the stress in a strip structure. Hu et al. [32] and Melville et al. [33] stacked GW signals from different excitation-sensing paths as an image. Then the CNN was utilized to process these images to localize simulated damages in the pressure vessel or an aluminum plate. In these studies, the CNN is preferred due to its unique structures such as local connection, shared weights and subsampling, allowing it to automatically extract representations of GW signals that are beneficial for damage diagnosis.

Instead of organizing damage indexes or stacking original signals, the time–frequency spectrogram (TFS) [34] of the GW signal has been more preferred as the input of the CNN model, since the TFS contains both the time-domain information and frequency-domain information. Besides, TFS is naturally an image that is more suitable to be processed with the CNN. Liu et al. [35] adopted the GW time–frequency image, which was obtained with short-time Fourier transform, to detect notched crack damage in an aluminum plate with the VGG-16 network. Compared to the short-time Fourier transform, the wavelet transform is a more powerful mathematical tool used to analyze the signal at different resolutions for nonperiodic and transient signals. Ewald et al. [36] and Rautela et al. [37] simulated GW signals in the notched plate. The wavelet coefficients of the GW signals were used as the CNN inputs for notch detection. Ebrahimkhanlou et al. [38] incorporated the CNN to localize the source of acoustic emission simulated with pencil break in an aluminum plate, in which wavelet coefficients of GW signals are normalized and converted to the input image. Wu et al. [39] adopted the TFS from GW signals in the composite plate to localize ply delamination with a deep CNN. Li et al. [40] used the synchrosqueezed wavelet transform to create the time–frequency image of acoustic emission wave data. Then multibranch CNN was combined for detecting cracks in the rail. All the studies show the effectiveness of the CNN-based damage diagnosis with TFS. However, most of them were performed on simulated damages in simple plate structures. Real fatigue cracks were rarely discussed. For engineering structures with real fatigue cracks, fatigue crack evaluation is affected by complicated uncertainties, from sources like crack geometries, holes, boundaries, and connections. These uncertainties make it difficult to extract GW features for the accurate diagnosis of the crack size, since changes of the GW signals caused by the fatigue crack may be masked. More importantly, the training data for real fatigue cracks are difficult to collect, resulting in a few-shot learning problem.

Aiming at automatically extracting more effective features from GW signals for evaluating fatigue crack size under complicated uncertainties, this paper proposes a fatigue crack evaluation method based on the GW–CNN ensemble and differential TFS. The GW signal is transformed into a two-dimensional TFS image with the complex Gaussian wavelet transform. The differential TFS between the baseline signal and the monitoring signal is processed as the CNN input in order to amplify the effect of the fatigue crack. Besides, an ensemble of CNNs is trained to determine the crack length in the structure jointly to deal with overfitting caused by small sample data. Each CNN in the ensemble automatically extracts high-level features from the TFS, aiming at evaluating the fatigue crack length rather than crack length levels.

The rest of this paper is organized as follows: the details of the proposed method are given in Section 2, including extracting the differential TFS with the complex Gaussian wavelet transform, as well as the construction of the GW–CNN ensemble. In Section 3, the fatigue tests of complicated lap joint structures are carried out, in which the GW-based technique is adopted for crack monitoring. The proposed method is then verified with the fatigue test data. Finally, the discussions and conclusions are in Section 4 and Section 5.

## 2. GW–CNN Ensemble-Based Fatigue Crack Evaluation Method

### 2.1. Differential Time–Frequency Spectrogram Extraction

GW is a kind of elastic wave that propagates in a wave-guide structure [41]. As shown in Figure 1, piezoelectric transducers (PZTs) are arranged or embedded in the structure. The GW signal is excited to propagate in the structure through the PZT and collected by the other PZT. If there is a fatigue crack, it would cause changes to the propagation of the GW signal. By comparing the monitored GW signal with the baseline signal obtained at the pristine structural state, the state of the fatigue crack can be identified.

Instead of directly extracting the damage index [11] from the original GW signals, this paper converts the GW signal into the TFS with the continuous wavelet transform to evaluate the crack length in the structure. The continuous wavelet transform [42] is a powerful tool to explore the characteristics of nonperiodic and transient signals such as GW signals in the time–frequency domain. It introduces an expandable spatial and frequency window called the “wavelet” which can overcome the inferiority of localization, characteristic in Fourier transform and short-time Fourier transform. In the time domain, a general wavelet transform can be expressed as follows,
(1)$$W_{\varphi }(a,b)=\frac{1}{\sqrt{a}}\int h(t)\varphi^{*}\left(\frac{t-b}{a}\right)dt,\;\;a>0$$
where *h*(*t*) represents the GW signal, *φ* represents the mother wavelet, *φ*^*^ is the complex conjugate of *φ*, *a* is the scaling factor that controls the wavelet’s frequency, and *b* is the translation factor that identifies its location.

The wavelet coefficients *W_φ_*(*a*,*b*) represent the similarity between the signal and the mother wavelet after being shifted and scaled. The scale of the wavelet can be converted to the frequency *f* = *f*_0 ·_
*f*_s/_*a*, where *f*_0_ is the center frequency of the mother wavelet, *f* is the instantaneous frequency, and *f*_s_ is the sampling rate. The selection of an appropriate wavelet is important, which directly affects the crack evaluation result. There are different kinds of wavelets, such as the Mexican Hat wavelet, the Morlet wavelet, and the Haar wavelet [43]. In this paper, the complex Gaussian wavelet is chosen as the best candidate due to its advantages of guaranteeing the edge position detected when changing the scale. The complex Gaussian wavelet is defined as Equation (2) [44]
(2)$$\varphi_{n}(t)=C_{n}\frac{d^{n}}{dt^{n}}(e^{-j\omega t}e^{-t^{2}})$$
where *n* denotes the order and *C_n_* is a normalizing constant that depends on *n*.

By using a series of wavelets with different scales, and shifting them in time along with the signal, a map of wavelet coefficients *W_φ_*(*a*,*b*) is obtained. The wavelet coefficients *W_φ_*(*a*,*b*) are complex values, and their modules can be arranged as an image, e.g., the TFS. In this paper, the differential TFS between the baseline signal and the monitoring signal is processed as the input of the CNN model, as expressed in Equation (3).
(3)$$\zeta (a,b)={\left(\left|W_{\varphi }(a,b)\right|\right)}^{2}-{\left(\left|W_{\varphi }^{\mathrm{baseline}}(a,b)\right|\right)}^{2}$$
where *ζ*(*a*,*b*) is called the differential magnitude. The square of the module is proposed to amplify the difference between the monitoring GW signal and the baseline.

### 2.2. GW–CNN-Based Crack Evaluation Model

The CNN is a well-known deep learning model which is quite suitable for processing images of classification and regression problems [45]. This subsection briefly introduces the underlying concept of the CNN.

As shown in Figure 2, the CNN usually has a multilayer structure, including the convolutional, activation, pooling, and full connection layer. The TFS image enters the network as the input, then the convolutional layers and pooling layers extract important local features. Finally, the crack length is outputted with the full connection layer.

The convolution layer utilizes the convolution operation to process the input image. In each convolutional layer, there are several convolutional kernels (or filters). The convolutional kernels consist of trainable weights which can generate a series of feature maps by sliding over the small local receptive fields of the input. The output of the *j*th feature map in the *l*th convolution layer is calculated as Equation (4) [40],
(4)$$x_{j}^{l}=f(\sum_{i\notin M_{j}}^{}x_{i}^{l-1}*w_{ij}^{l}+b_{j}^{l})$$
where $x_{i}^{l-1}$ denotes the *i*th feature map in the *l*-1th layer, $w_{ij}^{l}$ denotes the weight matrix of the *j*th filter, $b_{j}^{l}$ denotes the bias matrix of the *j*th filter, *M_j_* denotes the number of feature maps, * represents the convolution operation, and *f*(_·_) denotes the activation function.

The activation function *f*(_·_), named as the activation layer, brings nonlinearity to the CNN model by being imposed on the convolution filter output. Usually, the rectified linear unit (ReLU) activation function is preferred in deep learning because it has a simpler derivative result, leading to faster training and avoiding the gradient diffusion problem [46]. The expression of the ReLU activation function is given as Equation (5).
(5)$$\mathrm{ReLU}(x)=\left\{\begin{array}{l} 0\;\;\;\;\;\mathrm{if}\;\;x\leq 0\\ x\;\;\;\;\;\mathrm{if}\;\;x>0 \end{array}\right.$$

The pooling layer is commonly placed between successive convolution layers. It can progressively reduce the spatial size of the feature maps. This is also referred to as downsampling, by which the overfitting of the network can be controlled. Usually, downsampling can be implemented by operations like maximum pooling and average pooling. Assuming the pooling size is *c*, the average pooling feature of the *j*th region at the *l*th pooling layer is expressed as Equation (6).
(6)$$x_{j}^{l}=f(\beta_{j}^{l}\;\mathrm{mean}(x_{j}^{l-1})+b_{j}^{l})$$
where $\beta_{j}^{l}$ and $b_{j}^{l}$ represent the multiplicative and additive biases and mean(_·_) represents the average operation. The convolution layer detects the local connections of the features from the previous layer, while the pooling layer merges similar features and removes unnecessary irrelevant details.

The full connection layer is called the perceptron layer, which is applied at the end of the CNN to flatten the learned feature maps into one vector and obtain the expected output form. In this paper, the fatigue crack evaluation problem is a regression problem. Therefore, a linear activation function is adopted in the full connection layer, by which a scalar value denoting the crack length is outputted.

Moreover, there are auxiliary layers in the CNN, like the batch normalization layer and the dropout layer. The batch normalization layer normalizes a mini-batch of data across all training data for each channel independently, which can speed up training and reduce the sensitivity to network initialization. The dropout layer randomly sets input elements to zero with a given probability during training to deactivate a part of neurons on a particular layer [47]. This improves the generalization ability of the CNN because it forces the layer to learn with different neurons the same “concept”. During the prediction phase, the dropout is deactivated.

The training of CNN is a procedure of optimizing the weights and bias in the layers, via minimizing the loss function. In this paper, the mean square error is adopted as the loss function in accordance with the regression problem. For CNN optimization, the most popular algorithms include the stochastic gradient descent (SGD) and the adaptive moment estimation (Adam) algorithm [48]. The SGD is preferred for computer vision, while Adam is preferred for natural language processing and speech problems [40]. Therefore, the SGD algorithm is adopted in this paper. In the stochastic gradient descent algorithm, weights are updated after every sample is shown to the network. In order to speed up the training process, weights are updated only after a batch of images are shown to the network, named as the mini-batch SGD algorithm [49]. The training of the CNN is finished when the loss function is close to converging or a defined maximum number of training iterations is reached.

### 2.3. Fatigue Crack Evaluation Method Based on the GW–CNN Ensemble

As mentioned in Section 2.1, the GW signal collected at the structural pristine state is called the baseline signal. When a new GW signal is collected, its wavelet coefficients can be evaluated with the wavelet transform, along with the wavelet coefficients of the baseline signal. Then, the TFS image of this GW signal is evaluated with Equation (3). With the CNN mentioned above, the TFS image can be converted to the crack length in the structure, e.g., the fatigue crack evaluation, after the CNN is trained. Usually, the training dataset is collected by conducting a series of fatigue tests of several structures that are identical to the target structure. Due to the complexity of implementing fatigue tests, the number of the training structures are usually small. Therefore, the training dataset for the fatigue crack evaluation is a small sample problem. Besides, the training of the CNN includes several stochastic steps such as the parameter initialization. The CNNs will have different outputs even while being trained with the same dataset.

Therefore, this paper proposes the CNN ensemble for the fatigue crack evaluation problem, as shown in Figure 3. The CNN ensemble is composed of *M* CNNs, which are trained with the same dataset and settings. Assuming the *r*th CNNs outputs the crack length *y^r^* in the target structure, the final crack evaluation result is determined by all the CNNs’ outputs, as shown in Equation (7).
(7)$$\bar{y}=\frac{\sum_{r=1}^{M}y^{r}}{M}$$
where $\bar{y}$ is the crack evaluation result.

In general, the proposed GW–CNN ensemble-based fatigue crack evaluation method is implemented, as in Figure 3. At the structure’s pristine state, a GW signal is collected as the baseline. This GW signal is converted to the TFS with the continuous wavelet transform. During the service of the structure, the monitoring GW signal is acquired and used for fatigue crack evaluation. The differential TFS of the monitoring GW signal as Equation (3) is evaluated as the input of the CNN ensemble. By synthesizing the result of a series of CNNs, the final fatigue crack evaluation result is obtained.

## 3. Experimental Validation

The proposed method is validated on the fatigue test data of a lap joint structure, which is an important joint type of aircraft fuselage. The following firstly introduces the fatigue test settings, as well as some typical fatigue test results. Then, the training of the GW–CNN ensemble and its validation is carried out with the fatigue test data.

### 3.1. Fatigue Test Settings

As shown in Figure 4, the lap joint structure is made of the 2 mm thick 2024-T4 aluminum alloy plate, which is jointed by overlapping two plates with six countersunk rivets. It was designed and purchased by the authors, which was fabricated in a machining factory. According to the finite element analysis result, fatigue cracks are prone to initiate and grow at Nos. 4, 5, and 6 rivets due to the weakening of the countersunk holes. Therefore, two PZTs are arranged to monitor the growth of fatigue cracks, denoted as PZT1 and PZT2. Figure 5 shows the real specimen with the PZT smart layer designed by the authors’ group.

In Figure 6, the lap joint specimen is clamped vertically on the SUNS10T fatigue loading system. Since there is no precrack in the structure, a series of sinusoidal loading cycles are applied to initialize the fatigue crack at first. This load includes 13,000 sinusoidal cycles with a peak value of 25 kN and stress ratio of 0.1, and a number of sinusoidal cycles with a peak value of 20 kN and stress ratio of 0.1, until a crack is observed at any rivet hole. After that, a weakened load with a peak value of 15 kN and stress ratio of 0.1 is applied to slow the fatigue crack growth. This fatigue load was selected based on a previous study for this kind of lap joint. Firstly, finite element simulations were carried out to calculate the stress distribution in the structure, which is used to estimate its crack initiation with S-N curves. Then, previous fatigue tests of these kinds of lap joints were performed to adjust the fatigue load, to make the time duration of the crack initiation within several hours so to accelerate the fatigue test.

During the fatigue crack growth, the fatigue crack is observed through a digital microscope and the fatigue crack length is measured with scales on the specimen. The microscope model used in the fatigue test is BL-SM210, which has a maximum resolution of 210 M pixels, and 40 times zoom. The multichannel PZT array scanning system developed by the authors’ group [50] is used for the excitation and acquisition of GW signals. The excitation waveform is the three-cycle Hanning-windowed sine burst with the central frequency of 160 kHz. The GW signal is collected with a sampling rate of 10 MHz. Once the crack grows for a certain length, the fatigue load is suspended and a static load of 5 kN is applied. Meanwhile, the GW signal is excited and acquired with PZT1 as the actuator and PZT2 as the sensor. In total, six specimens, labeled from D1 to D6, are involved in the fatigue test. Before the crack growth of each specimen, a baseline GW signal is obtained at the pristine state.

### 3.2. Fatigue Test Results of the Lap Joint Structure

Figure 7 illustrates the fatigue cracks in the lap joint specimen D2. After a number of loading cycles, the fatigue crack occurs at the edge of the No. 4 rivet. The crack continuously grows and another crack initiates at the edge of No. 5 rivet. Finally, the two cracks link together. The locations and geometries of the cracks in the other specimens are different from Figure 7. To simplify the data processing, the total crack length of these cracks is considered in this paper. Figure 8 shows the total crack length versus the loading cycles recorded for the specimens mentioned above.

Typical guided wave signals are given in Figure 9. For each GW signal, the front wave packet is the crosstalk that is useless and introduced by the electromagnetic induction between the circuits of the actuator and sensor. Excluding the crosstalk, the direct wave packet is usually used for crack monitoring since signals after this packet are affected by different kinds of uncertainties and show large dispersion between different specimens. It can be found the shape of the direct wave packet is different from the excitation signal. This is due to frequency dispersion, as well as reflections and scattering caused by the rivets and boundaries. Small changes can be observed in the direct wave packet with the crack growth.

### 3.3. GW–CNN Ensemble-Based Fatigue Crack Evaluation

In this paper, the time–frequency spectrogram, e.g., TFS, is evaluated with the complex Gaussian wavelet transform. Figure 10 shows the TFS extracted from the direct wave packet at different crack lengths in specimen D1. The frequency of the wavelet is chosen from 100 kHz to 300 kHz with an interval of 1 kHz. As a result, the TFS has a size of 201 × 213. The TFS clearly shows the time–frequency distribution of the direct wave packet. It can be found there is an increase of the magnitude from Figure 10a–f. This is because the reflection wave from the crack will increase when the cracks grow, leading to an increase of the signal amplitude of the direct wave packet.

After that, the differential TFS is evaluated as the input of the GW–CNN ensemble, as shown in Figure 11. At the initial time, there is no crack length, thus the image is zero. With the growth of the cracks, the magnitude in the TFS increases, which locates at the tail of the direct wave packet. This is because the reflection of the GW signal from the crack has a longer propagation distance than the wave directly from PZT1 to PZT2. The differential TFS well represents the growth of the fatigue cracks.

Aiming at verifying the GW–CNN ensemble-based fatigue crack evaluation method on the lap joint structure, the CNN is firstly designed as Figure 12. There are five convolution layers and three pooling layers. In each convolution layer, the ReLU activation function is used. After the convolution operation, batch normalization is conducted. A dropout layer is added before the full connection layer. The output of the CNN is the total crack length in the lap joint structure, whose dimension is 1. To construct the CNN ensemble, a total of *M* = 20 CNNs with the same structure are employed.

In order to train the CNN, the data from the former specimens D1–D5 are used as the training samples. Besides, the data of specimen D6 are used for verifying the proposed method. Firstly, the GW signals collected at different crack lengths in specimens D1–D5 are used to evaluate corresponding differential TFS images, resulting in a dataset size of 77. That is, a total of 77 TFS images are used for training. In addition, there are 17 images obtained from specimen D6, which are used for testing the proposed model. There is no postprocessing for the TFS images. The training dataset is normalized to train the 20 CNNs with the same settings. The training algorithm is the mini-batch SGD algorithm. The initial learning rate is 1 × 10^−4^, which will decrease for every 200 epochs with a rate of 0.5. The batch size is 8 and the maximum number of epochs is 1000. The computer used for training runs on the Intel(R) Core(TM) i7-10750H CPU, GeForce GTX1660ti GPU, and 16 GB RAM.

## 4. Results and Discussions

Figure 13 shows the typical loss varying with the training process. No validation data is used during the training process, since only 77 TFS images are collected from specimens D1–D5 for training. It can be found that the loss converges after sufficient iterations. The time for training each CNN is about 9 min. The root mean square error (RMSE) of the trained dataset is 1.4 mm.

After the training of the GW–CNN ensemble, the TFS image from specimen D6 is sequentially evaluated and input into the GW-CNN ensemble, which outputs the evaluation result of the crack length. As shown in Figure 14a, there are 20 crack length outputs at each loading cycle, which are evaluated with the 20 CNNs in the GW–CNN ensemble, respectively. The outputs of the CNNs show large dispersion, since the number of the training TFS images is small. In addition, Figure 14b illustrates the final crack evaluation results by averaging the CNNs’ outputs with Equation (7). The actual cracks of specimen D6 are given in Figure 15. The crack firstly initiates at the right hole edge of the No. 4 rivet and grows with fatigue loading. Then, new cracks occur at another hole edge of the No. 4 rivet and the No. 5 rivet. The total crack length of these cracks is evaluated with the trained GW–CNN ensemble, which coincides with the experimental crack length.

The commonly used damage index-based crack evaluation method is also considered for comparison. In this paper, two kinds of widely used damage indexes are extracted from the guided wave signal, which are the root mean square deviation damage index [51] and the drop-in correlation coefficient damage index [52]. The root mean square deviation damage index mainly represents the energy change of the GW signal. The drop-in correlation coefficient damage index mainly represents the phase changes of the GW signal. The direct wave packet in the GW signal, as shown in Figure 9b, is intercepted for evaluating the damage index since it is less affected by the boundaries of the structure. The extracted damage indexes of specimens D1–D6 are given in Figure 16. It can be found that the damage indexes for the different specimens show large dispersion. This is due to the geometric complexity of the lap joint structure and the uncertainties of fatigue cracks. The lap joint structure is composed of two aluminum plates which are assembled by the rivets. The guided wave propagating in the structure is not only affected by the fatigue cracks, but also by the rivet holes, the rivets, and the contacts between the two plates. These effects are different and uncertain for different specimens. The geometries and locations of the fatigue cracks in different specimens are different due to the stochastic nature of fatigue crack initiation and growth. Guided wave signals acquired under the same crack length from different specimens are different. The above-mentioned problems lead to the large dispersion of the damage indexes.

The damage index-based crack evaluation needs a surrogate model to establish the mapping between the damage index vector and the crack length. In this paper, a two-input and two-order polynomial mapping is also constructed with the data from specimens D1–D5. The data from specimen D6 are used for testing the damage index-based crack evaluation model. The result is shown in Figure 17.

There are large errors between the crack evaluation results obtained with the damage index-based method. Figure 18a shows the absolute errors of the GW–CNN ensemble and the damage index-based method. The horizontal axis is the experimental crack length. The fatigue evaluation result of the damage index-based method shows small errors when the crack length is small but expresses large errors when the crack length is large. This is because the damage index-based model represents the average relationship of specimens D1–D5, which deviates from the new target specimen D6. This deviation is bigger when the crack length is larger. On the other hand, the proposed GW–CNN ensemble shows smaller errors than the damage index-based method, especially when the crack length is large. The maximum evaluation error with the proposed GW–CNN ensemble is 3 mm, which occurs when the actual crack length is 5 mm. This point is singular, which may be caused by the training data. Figure 18b gives the relative errors of the evaluation results. The relative error of the GW–CNN ensemble is large when the crack length is small. However, it is acceptable if we consider the complexities of the lap joint structure and the absolute errors. The effect of a small fatigue crack on the GW propagation may be masked by the effects of the rivets, holes, and contacts. It is difficult to have a very small relative error when the crack length is small. The RMSE of the GW–CNN ensemble is 1.7 mm, compared with 12.7 mm of the damage index-based method. In summary, the proposed GW–CNN method gives satisfying results, thus verifying the effectiveness of the proposed method. The proposed method offers a new way for automatically extracting more effective features from GW signals aiming at evaluating real fatigue cracks under complicated uncertainties.

## 5. Conclusions

This paper proposes a GW–CNN ensemble-based method for the on-line evaluation of fatigue cracks under complicated uncertainties. The GW signal is transformed into a two-dimensional TFS image with the complex Gaussian wavelet transform, by which the differential TFS between the baseline signal and the monitoring signal is processed as the CNN input. Besides, an ensemble of CNNs is trained to jointly determine the crack length in the structure. The fatigue tests of complicated lap joint structures are carried out to validate the proposed method. The data of the former five specimens, D1–D5, are used to train the GW–CNN ensemble, and the data collected from specimen D6 is used for testing. The crack evaluation result shows the proposed ensemble can effectively diagnose the total crack length in the lap-joint structure compared with the traditional damage index-based method. The maximum evaluation error with the proposed GW–CNN ensemble is 3 mm and the root mean square error is 1.7 mm, showing the effectiveness of extracting more effective features from the TFS with the CNN.

The proposed method is a kind of data-driven method which depends on the training data. The more data is collected, the better the method performs. However, carrying out fatigue tests of the structure to collect the training data is time and cost consuming, which is the main limitation of the method. Therefore, in the future, simulation data of GW signals with the finite element method can be a useful complement for enriching the training dataset. Moreover, real fatigue tests, including different load levels, a different disposition of the rivets, etc., may also be considered for correcting the finite element model, thus aiming at providing more effective data.

Bedsides, all the acronyms are listed in Appendix A to make them clear.

## Figures and Tables

**Figure 1 sensors-22-00307-f001:**
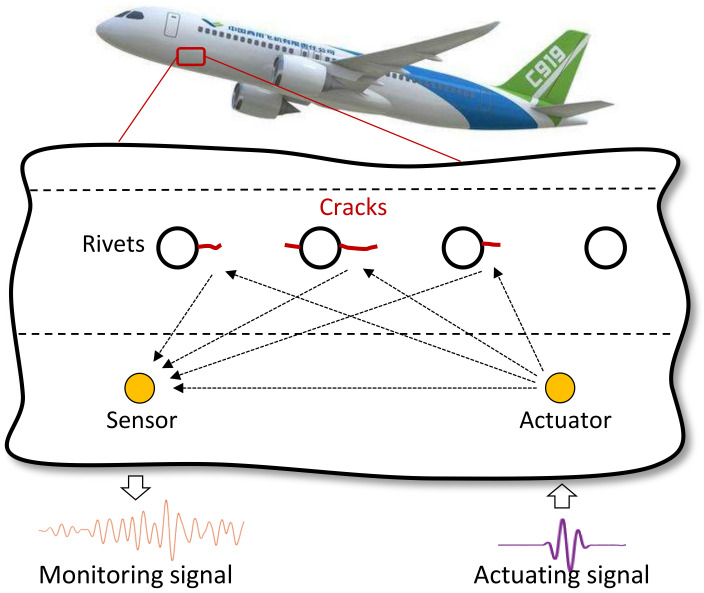
Illustration of the GW-based crack monitoring.

**Figure 2 sensors-22-00307-f002:**
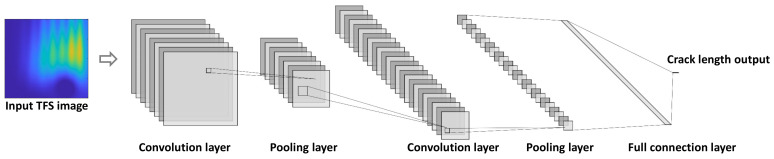
Architecture of a typical CNN.

**Figure 3 sensors-22-00307-f003:**
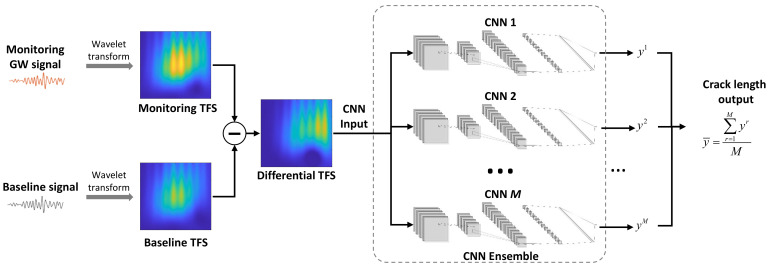
Implementation of the GW–CNN ensemble-based fatigue crack evaluation.

**Figure 4 sensors-22-00307-f004:**
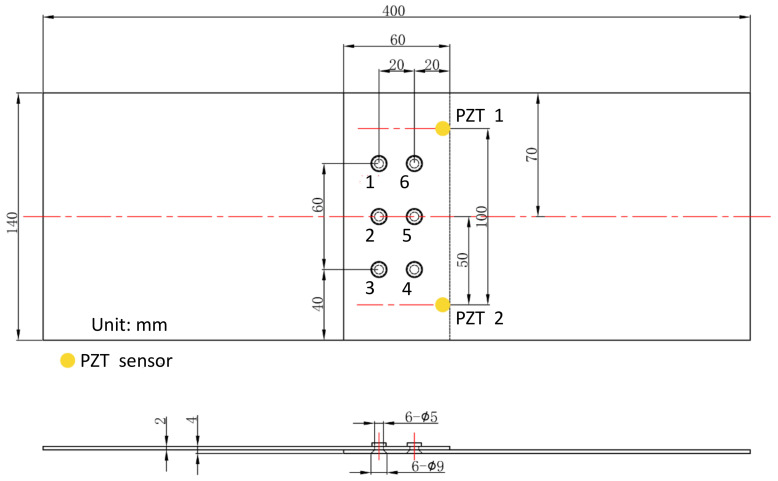
Specimen dimensions and sensor layout.

**Figure 5 sensors-22-00307-f005:**
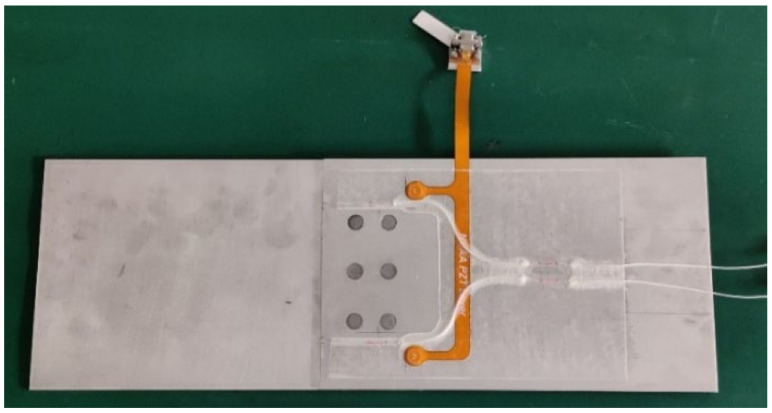
Specimen with the PZT smart layer.

**Figure 6 sensors-22-00307-f006:**
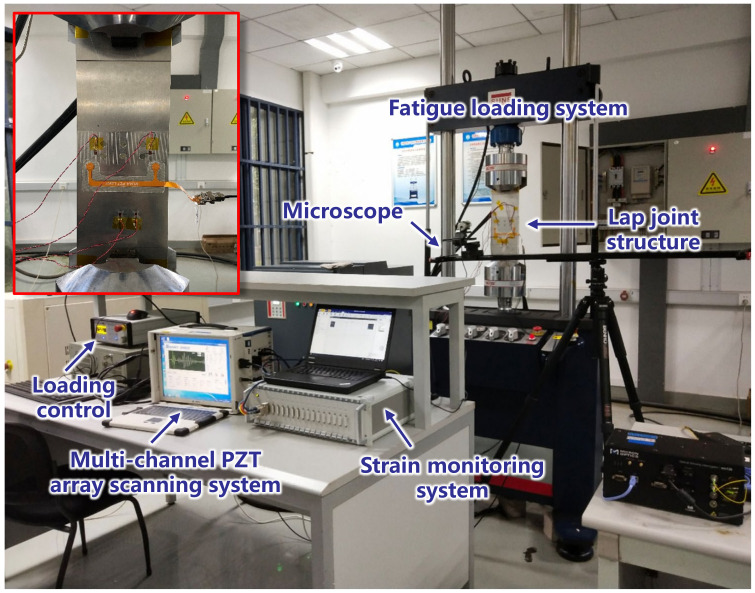
Setup of the fatigue test.

**Figure 7 sensors-22-00307-f007:**
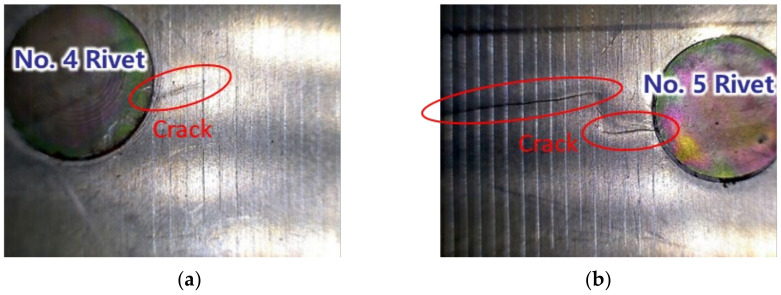
Fatigue cracks of the lap joint structure: (**a**) Fatigue crack just occurs at No. 4 rivet; (**b**) Cracks from No. 4 rivet and No. 5 rivet link together.

**Figure 8 sensors-22-00307-f008:**
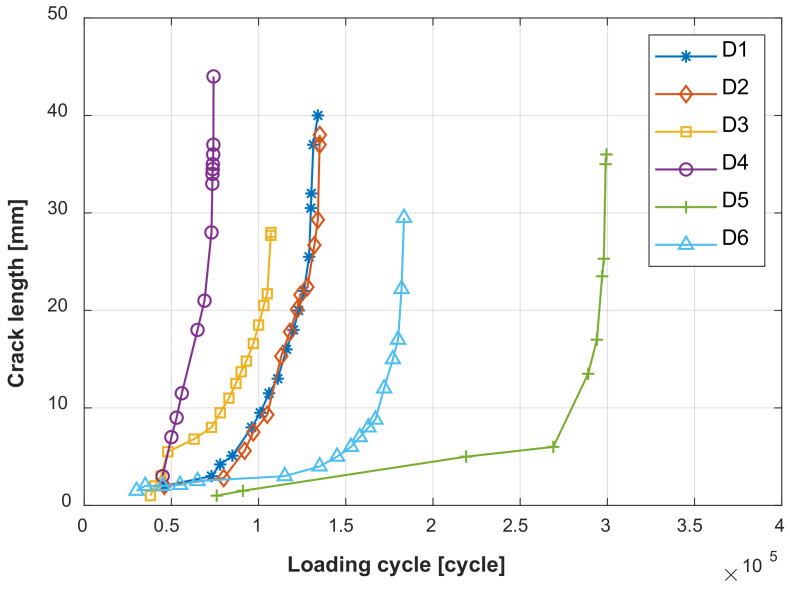
Total crack length versus fatigue loading cycles.

**Figure 9 sensors-22-00307-f009:**
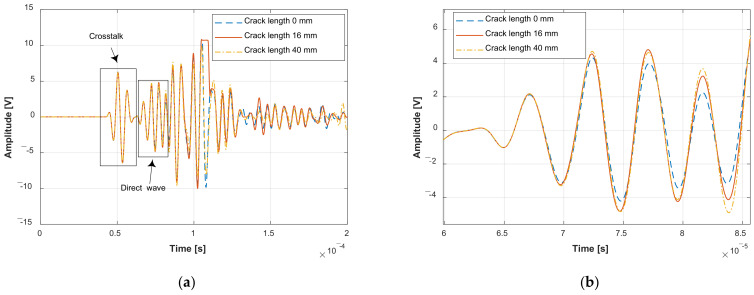
Typical guided wave signals with the increase of the crack lengths: (**a**) Whole guided wave signal collected during the fatigue test; (**b**) Magnification of the direct wave packet of the collected GW signal.

**Figure 10 sensors-22-00307-f010:**
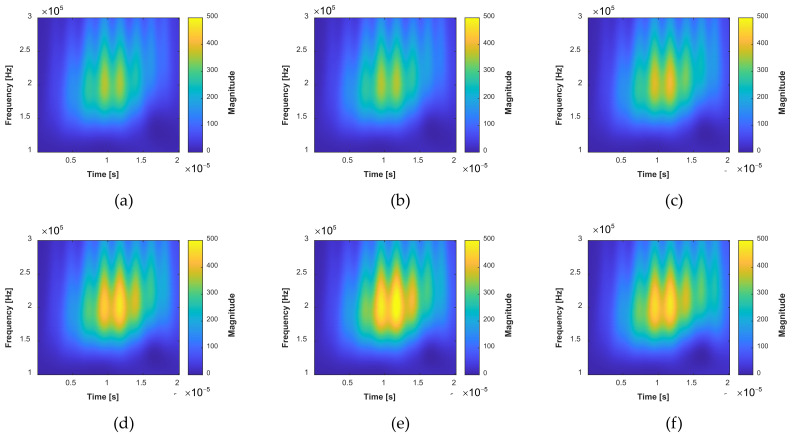
Extracted TFS images with the fatigue crack growth in specimen D1: (**a**) The TFS image of the GW signal collected when the total crack length is 0 mm obtained by the continuous wavelet transform; (**b**) The total crack length is 4.2 mm; (**c**) The total crack length is 9.5 mm; (**d**) The total crack length is 16 mm; (**e**) The total crack length is 22 mm; (**f**) The total crack length is 32 mm.

**Figure 11 sensors-22-00307-f011:**
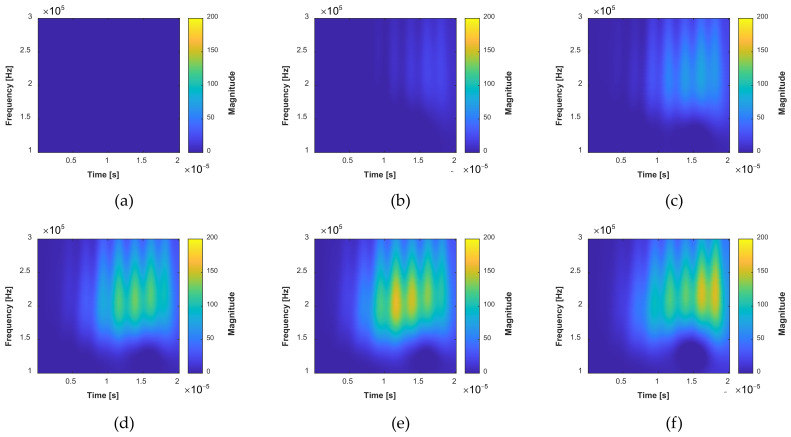
Extracted differential TFS images with the fatigue crack growth in specimen D1: (**a**) The total crack length is 0 mm obtained by the continuous wavelet transform; (**b**) The total crack length is 4.2 mm; (**c**) The total crack length is 9.5 mm; (**d**) The total crack length is 16 mm; (**e**) The total crack length is 22 mm; (**f**) The total crack length is 32 mm.

**Figure 12 sensors-22-00307-f012:**
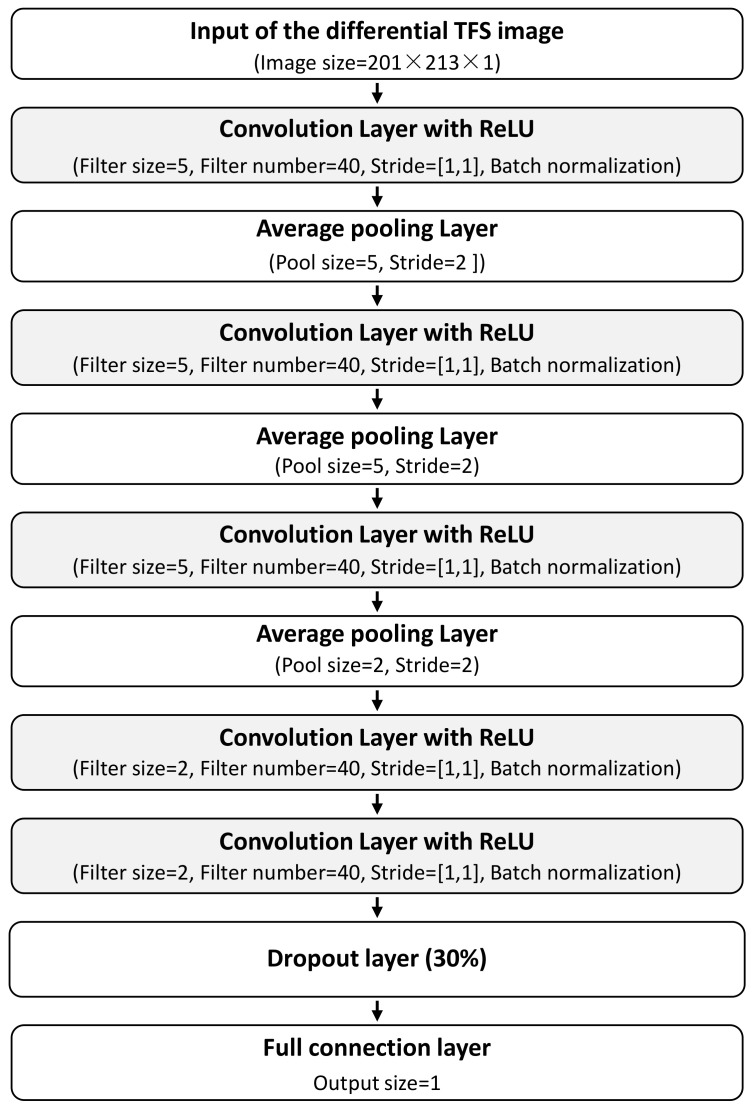
Structure of the CNN for fatigue crack evaluation.

**Figure 13 sensors-22-00307-f013:**
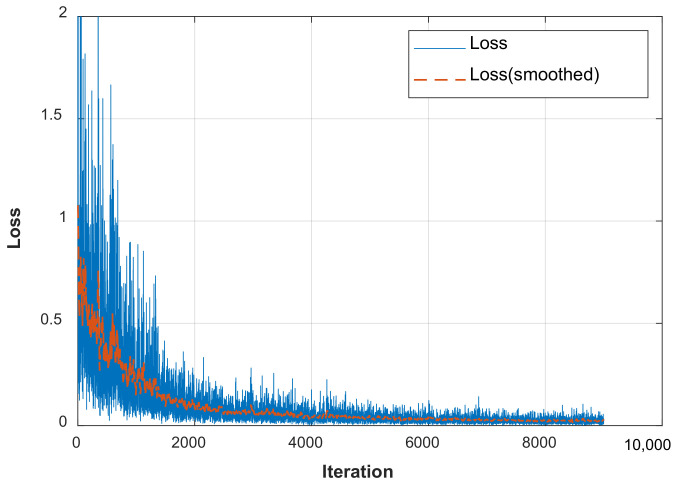
Loss varying with the training process.

**Figure 14 sensors-22-00307-f014:**
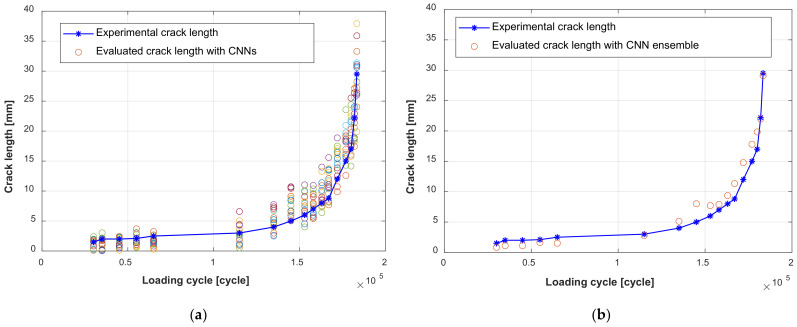
Fatigue crack evaluation result with the GW–CNN ensemble: (**a**) Output of the CNNs in the CNN ensemble; (**b**) Final output of the CNN ensemble.

**Figure 15 sensors-22-00307-f015:**
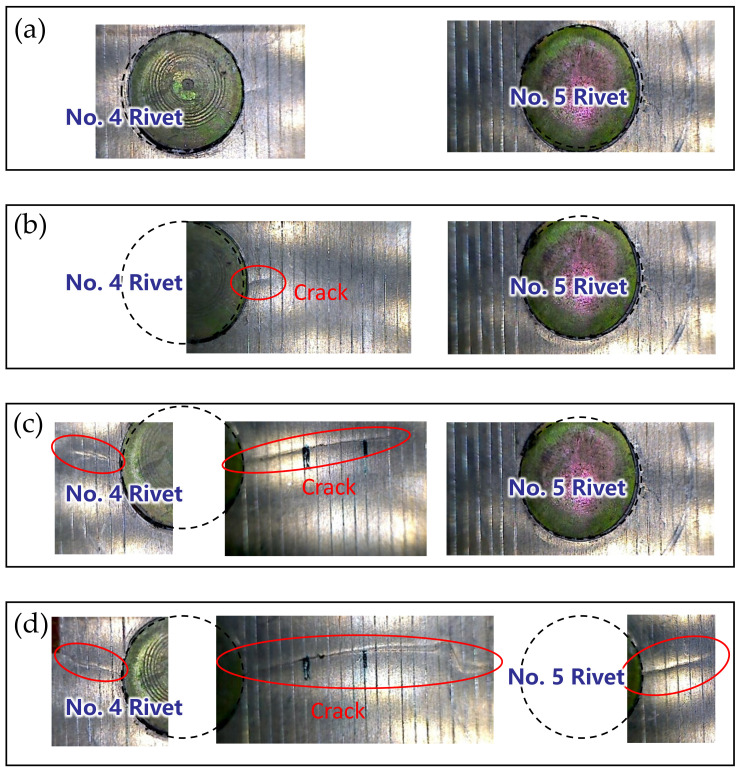
Fatigue cracks of specimen D6 at different stages: (**a**) Total crack length of 0 mm; (**b**) Total crack length of 2 mm; (**c**) Total crack length of 17 mm; (**d**) Total crack length of 29.5 mm.

**Figure 16 sensors-22-00307-f016:**
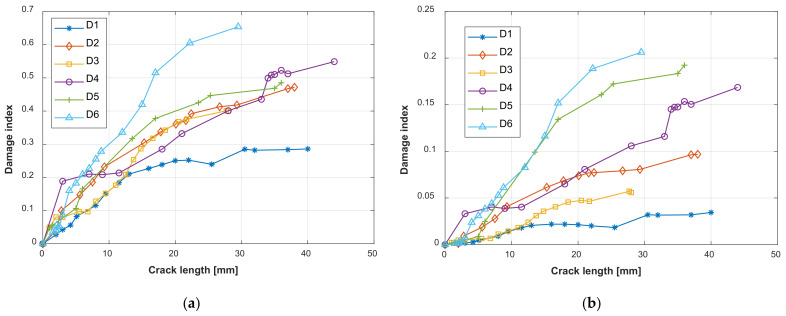
Widely used damage indexes extracted from the GW signal: (**a**) Root mean square deviation damage index; (**b**) Drop-in correlation coefficient damage index.

**Figure 17 sensors-22-00307-f017:**
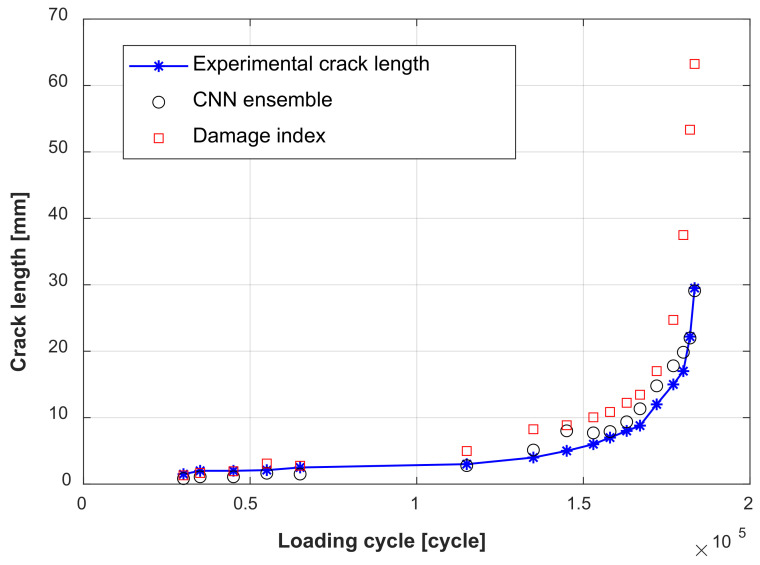
Fatigue crack evaluation result compared with the damage index-based method.

**Figure 18 sensors-22-00307-f018:**
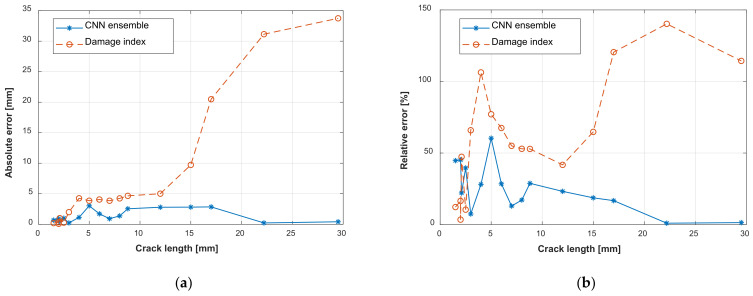
Fatigue crack evaluation errors: (**a**) Absolute error; (**b**) Relative error.

## Data Availability

Not applicable.

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
