# Peer review of "Fatigue Crack Evaluation with the Guided Wave–Convolutional Neural Network Ensemble and Differential Wavelet Spectrogram"

_sensors, 2021, doi:10.3390/s22010307_

Round 1
Reviewer 1 Report
In my opinion, the article entitled "Fatigue crack evaluation with the guided wave-convolutional neural network ensemble and differential wavelet spectrogram" is interesting and well written. However, some revisions might need to enhance the quality of the article. Some points I have indicated throughout the manuscript need to upgrade. Details of my comments are in the attached pdf file. Good luck to the authors.

Author Response
Thanks for your comments, the following is my point-by-point response:
(1) Page 1. “CNNs are trained” Need to mention the training and testing accuracy of the CNN model you used in this study.
Reply: The root mean square error (RMSE) of the training dataset is 1.4 mm. And the RMSE of the testing dataset is 1.7 mm with the CNN ensemble in this study. These results are added in the Abstract.
(2) Page 1. “is 1.7mm”, It should be in past tense right?
Reply: Thanks for your comments, the tense of the sentences of validation is revised.
(3) Page 1. “SHM has” Reference Needed
Reply: the Reference [6] is added according to the comment
(4) Page 2, the last paragraph. The objective of this work is not clear in this paper. Please make it clear in this section.
Reply: The end of the introduction is rewritten to make the objective of this work clearer.
(5) Using the method mentioned in this study, are you targeting to detect crack of a contact-less object from a certain distance? If so, up-to which distance you can detect the crack using this method? And what are the challenges you are anticipating to make this method work? Please specify
Reply: The method in this study does not work in a contact-less manner. In the guided wave-based method, piezoelectric transducers are adhered on the structure where cracks are possible to occur. One piezoelectric transducer is used to excite the elastic wave in the structure, called the guided wave. And another piezoelectric transducer is used to acquire the guided wave signal propagating in the structure. The main challenge is the geometric complexity of the lap joint structure. There are holes, boundaries, rivets, and contacts between the plates in the structure, which may also have effects on the guided wave signal except for fatigue cracks.
The above discussions are added in the Section 4.
(6) Fig. 2. Please specify the size of each layer in the figure
Reply: In Section 2.1, Fig. 2 is an illustration of a conventional convolutional neural network. Therefore, the size of each layer was not given in Fig. 2. The size of each layer is given in Fig. 12.
(7)Page 4. “Usually the ReLU activation function…”, Please provide reference to verify this statement
Reply: Reference [46] is added according to the comments.
(8)Page 5. “The SGD is preferred” Reference is needed here
Reply: Reference [40] is added according to the comments.
(9) Page 5. “Please explain the process how the TFS image obtained from the GW signals”
Reply: A description is added at the beginning of Section 2.3, to make the process of obtaining the TFS image clear.
(10)Page 10. Need to provide the image number used for training, the number of images used for validate the model. If any post processing (i. e., augmentation, distortion , etc.) used to build this model need to mention and describe
Reply: Total 77 images are used for training, and 17 images are used for testing the proposed method. There is no post processing for the images. These descriptions are added in Section 3.3
(11)Page 11. The accuracy of the model need to be mentioned clearly.
Reply: More discussions on the accuracy of the model is added in Section 4.
(12) Page 12. I think you need to describe more about the validation process and the findings of the application.
Reply: More discussions are given in Section 4.
(13)Page 12. What was the error (%) while the model used for detecting the crack length? Need to specify and describe clearly.
Reply: The relative errors are added in Fig. 18 of the revised manuscript, which are also discussed in Section 4.
(14) Page 13. What would be the benefit if anyone uses this model developed in this study? Need to mention the future benefit of this model according to the authors perspective.
Reply: The traditional damage index-based method and machine learning methods require manual processing and selection of GW features, which depend highly on expert knowledge and are easily affected by complicated uncertainties. The proposed method offered a new way for automatically extracting more effective features from GW signals aiming at evaluating real fatigue cracks under complicated uncertainties.
The above descriptions are added at the end of Section 4.
Reviewer 2 Report
This paper deals with the fatigue crack evaluation with a guided-wave-convolutional neural network approach. The topic is interesting, and the paper is well structured, however, there are some concerns that must be addressed prior its publication in Sensors.
- It would be very useful to include an acronym index to facilitate the paper reading.
- There is a lack of some information about the experimental procedure. For example, how were obtained the aluminum plates and the rives? Were they purchased? How were selected the values of the loads for crack initiation? Are they based on a previous study for this kind of lap joints?
- Please, include the microscope model and their maximum resolution.
- Do you have an idea of the resolution of the proposed technique? That is, the minimum expected crack length that you are able to detect.
- Which is the main advantage of this technique over the conventional one using a microscope?
- How do you explain the differences observed in Figure 16 between the damage index-based method and the experimental measurements?
- In my opinion, it would be helpful to include some images of the damaged specimen in several stages and their comparison to the measurements given by your proposed technique.
- I think that it would have been useful to expand the number of experiments, i.e. by including different load levels, a different disposition of the rivets, etc. If it is not possible for this study, maybe you should consider it for future research.
Author Response
Thanks for your comments, the following is my point-by-point response
(1) It would be very useful to include an acronym index to facilitate the paper reading.
Reply: Thanks for your suggestion. An acronym table is added in the Appendix at the end of the revised manuscript.
(2)There is a lack of some information about the experimental procedure. For example, how were obtained the aluminum plates and the rivets? Were they purchased? How were selected the values of the loads for crack initiation? Are they based on a previous study for this kind of lap joints?
Reply:
(a)Yes, this lap joint structure was designed and purchased by the authors, and was fabricated in a machining factory.
(b)Yes, this fatigue load was selected based on a previous study for this kind of lap joint structure. Firstly, finite element simulations were carried out to calculate the stress distribution in the structure, which is used to estimate its crack initiation with S-N curves. Then, previous fatigue tests of this kind of lap joints were performed to adjust the fatigue load, to make the time duration of crack initiation within several hours so that accelerating the fatigue test.
The above descriptions are added in the revised manuscript.
(3)Please, include the microscope model and their maximum resolution.
Reply: The microscope model used in the fatigue test is BL-SM210, which has a maximum resolution of 210M pixels and 40 times zoom.
This information is added in the revised manuscript.
(4) Do you have an idea of the resolution of the proposed technique? That is, the minimum expected crack length that you are able to detect.
Reply: The minimum expected crack length that is able to detect is about 2 mm.
(5) Which is the main advantage of this technique over the conventional one using a microscope?
Reply: In practical engineering, the main advantage of the proposed guided wave and convolutional neural network-based structural health monitoring (SHM) technique is that it can be performed in any time after the sensors are arranged on the structure, so that it can examine the state of fatigue cracks in real-time to demine whether or when maintenance operations are needed. Otherwise, the conventional microscope-based method can only be carried out off-lien which is time and labor consuming, or even infeasible when the target part is inaccessible for a microscope.
(6) How do you explain the differences observed in Figure 16 between the damage index-based method and the experimental measurements?
Reply: This is due to the geometric complexity of the lap joint structure and uncertainties of fatigue cracks:
(a) The lap joint structure is composed of two aluminum plates, which is assembled by the rivets. The guided wave propagating in the structure is not only affected by the fatigue cracks, but also by the rivet holes, the rivets, the contacts between the two plates, and sensor properties. These effects are different and uncertain for different specimens.
(b) The geometries and locations of the fatigue cracks in different specimens are different due to the stochastic nature of fatigue crack initiation and growth. Guided wave signals acquired under the same crack length from different specimens are different.
The above-mentioned two problems lead to the large dispersion of the damage indexes.
And the damage index-based model represents the average relationship of specimen D1-D5, which deviates from the new target specimen D6. This deviation is bigger when the crack length is larger. Therefore there are differences between the damage index-based method and the experimental measurements
These discussions are added in Section 4.
(7) In my opinion, it would be helpful to include some images of the damaged specimen in several stages and their comparison to the measurements given by your proposed technique.
Reply: Thanks for your suggestion, the actual crack images are added as Fig. 15 and compared in Section 4.
(8) I think that it would have been useful to expand the number of experiments, i.e. by including different load levels, a different disposition of the rivets, etc. If it is not possible for this study, maybe you should consider it for future research.
Reply: Thanks for your suggestions. You are correct that the more data is obtained, the better performance the method will have. However, designing and carrying out fatigue tests sometimes need months of works. The authors will consider your suggestions in the future.
The above discussions are added in the Conclusions.
Reviewer 3 Report
Reviewer’s Report on the manuscript entitled:
Fatigue crack evaluation with the guided wave-convolutional neural network ensemble and differential wavelet spectrogram
The authors proposed a fatigue crack evaluation framework with the GW-convolutional neural network ensemble and differential wavelet spectrogram. In my view, the method and results are interesting and promising. I have a few minor suggestions listed below:
Line 16. Please define CNN. Please check all the acronyms used in the manuscript and define them the first time they are used. They must be defined both in the Abstract and the Manuscript. Please use a consistent style when defining them. Also, please add an acronym table at the end of the manuscript listing all the acronyms used in the paper.
Line 28. Please replace “of considerable significance” with “crucial”
Line 52. Please rewrite this sentence to sound more professional.
Lines 71, 76. Please add this review article here describing the time-frequency spectrograms and state-of-the-art time-frequency analysis methods: https://doi.org/10.3390/app11136141
Line 133. It should be Haar, not Harr
Lines 141, 343, etc. It should be “…, e.g., …”
Line 200. Please add the following reference:
Kingma, D. P.; Ba, J.L. Adam: A Method for Stochastic Optimization. arXiv 2017, arXiv:1412.6980. Available online: https://arxiv.org/abs/1412.6980 (accessed on 15 December 2021).
Line 240. Please replace the dot with a comma.
In the Conclusions, please also mention the limitations and future direction of this study.
Thank you for your contribution
Regards,
Author Response
Thanks for your comments, the following is my point-by-point response:
(1) Line 16. Please define CNN. Please check all the acronyms used in the manuscript and define them the first time they are used. They must be defined both in the Abstract and the Manuscript. Please use a consistent style when defining them. Also, please add an acronym table at the end of the manuscript listing all the acronyms used in the paper.
Reply: Thanks for your comments. The authors checked the acronyms. And an acronym table is added at the end of the revised manuscript.
(2) Line 28. Please replace “of considerable significance” with “crucial”
Reply: Revised as the comment
(3) Line 52. Please rewrite this sentence to sound more professional.
Reply: The sentence is rewritten as “Recently, deep learning (DL) has already made a huge impact in areas, such as cancer diagnosis, precision medicine, self-driving cars, predictive forecasting, and speech recognition.”
(4) Lines 71, 76. Please add this review article here describing the time-frequency spectrograms and state-of-the-art time-frequency analysis methods: https://doi.org/10.3390/app11136141
Reply: The reference [34] is added in the revised manuscript.
(5) Line 133. It should be Haar, not Harr
Reply: Thanks, it is revised.
(6) Lines 141, 343, etc. It should be “…, e.g., …”
Reply: The authors checked the whole manuscripts, and it is revised.
(7) Line 200. Please add the following reference:
Kingma, D. P.; Ba, J.L. Adam: A Method for Stochastic Optimization. arXiv 2017, arXiv:1412.6980. Available online: https://arxiv.org/abs/1412.6980 (accessed on 15 December 2021).
Reply: The reference [48] is added.
(8) Line 240. Please replace the dot with a comma.
Reply: Revised as the comment
(9) In the Conclusions, please also mention the limitations and future direction of this study.
Reply: The proposed method is a kind of data-driven method, which depends on the training data. The more data is collected, the better the method performs. However carrying out fatigue tests of the structure to collect the training data is time and cost consuming, which is the main limitation of the method. Therefore, in the future, simulation data of GW signals with the finite element method can be useful complements for enriching the training dataset. Also, real fatigue tests including different load levels, a different disposition of the rivets, etc. are also considered for correcting the finite element model, aiming at providing more effective data.
The above discussions are added in the Conclusions.